# Cytosine Methylation in Genomic DNA and Characterization of DNA Methylases and Demethylases and Their Expression Profiles in Viroid-Infected Hop Plants (*Humulus lupulus* Var. ‘Celeia’)

**DOI:** 10.3390/cells11162592

**Published:** 2022-08-19

**Authors:** Andrej Sečnik, Nataša Štajner, Sebastjan Radišek, Urban Kunej, Mitja Križman, Jernej Jakše

**Affiliations:** 1Department of Agronomy, Biotechnical Faculty, University of Ljubljana, 1000 Ljubljana, Slovenia; 2Plant Protection Department, Slovenian Institute of Hop Research and Brewing, 3310 Žalec, Slovenia; 3Laboratory for Food Chemistry, National Institute of Chemistry, Hajdrihova 19, 1000 Ljubljana, Slovenia

**Keywords:** DNA methylation, viroid, hop plant, HPLC, 5-mC, DNA methylase, DNA demethylase, phylogeny, RNA-directed DNA methylation, RT-qPCR

## Abstract

Abiotic and biotic stresses can lead to changes in host DNA methylation, which in plants is also mediated by an RNA-directed DNA methylation mechanism. Infections with viroids have been shown to affect DNA methylation dynamics in different plant hosts. The aim of our research was to determine the content of 5-methylcytosine (5-mC) in genomic DNA at the whole genome level of hop plants (*Humulus lupulus* Var. ‘Celeia’) infected with different viroids and their combinations and to analyse the expression of the selected genes to improve our understanding of DNA methylation dynamics in plant-viroid systems. The adapted HPLC-UV method used proved to be suitable for this purpose, and thus we were able to estimate for the first time that the cytosine methylation level in viroid-free hop plants was 26.7%. Interestingly, the observed 5-mC level was the lowest in hop plants infected simultaneously with CBCVd, HLVd and HSVd (23.7%), whereas the highest level was observed in plants infected with HLVd (31.4%). In addition, we identified three DNA methylases and one DNA demethylase gene in the hop’s draft genome. The RT-qPCR revealed upregulation of all newly identified genes in hop plants infected with all three viroids, while no altered expression was observed in any of the other hop plants tested, except for CBCVd-infected hop plants, in which one DNA methylase was also upregulated.

## 1. Introduction

DNA methylation is one of the best studied epigenetic modifications [1] and plays an important role in plant responses to various biotic and abiotic environmental stimuli [2]. The DNA methylation landscape is tremendously dynamic. It differs from plant species to plant species and depends on the type of tissue [3], the changes can be reversible or constant and are transmitted during meiosis and mitosis [4,5,6], the methylation pattern changes during plant development [7]. Finally, changes in DNA methylation can be induced by abiotic and biotic stresses [8,9,10,11,12,13]. DNA methylation is catalysed by DNA methyltransferases and occurs by the addition of a methyl group to the C-5 site of cytosine, the N-6 site of adenine, and the N-7 site of guanine [14]. It is also regulated by the DNA demethylases, which can excise 5-mC from sequence contexts [15]. DNA methylation in plants is known to occur frequently at cytosine bases in the symmetric context (CG and CHG) and in the asymmetric context (CHH) [16]. Previous studies have shown that DNA methylation of cytosine at the C-5 site is important for several biological processes, including genome stability, gene imprinting, growth and development, stress response, and biosynthetic regulation of secondary metabolites [17,18,19].

Viroids are the least complex infectious agents known. They consist entirely of a tiny, single-stranded, circular RNA with a strong secondary structure. Unlike viruses, their genome, which comprises about 246 to 401 nucleotides, is non-translatable [20]. Many economically important crops such as apples, avocados, coconuts, grapevines, hop plants, peaches, potatoes, and tomatoes can develop disease symptoms caused by viroids [21]. The non-coding nature suggests that the RNA of viroids must redirect the host machinery through the RNA molecule itself to allow infection and replication. The hop plant (*Humulus lupulus*) is an important crop grown primarily for its use in the brewing industry. Its production is threatened by diseases that can significantly reduce yields. Apple fruit crinkle viroid (AFCVd), citrus bark cracking viroid (CBCVd), hop latent viroid (HLVd) and hop stunt viroid (HSVd) are four different viroids that can infect hop plants. There is growing evidence of dynamic changes in host DNA methylation as a result of viroid-induced RNA-directed DNA methylation (RdDM) during infection [22,23,24,25,26,27]. Based on the promising results of these studies, it is becoming increasingly clear that viroid-derived small interfering RNAs (vd-siRNAs) interfere with the RdDM machinery and are therefore closely associated with changes in the methylation profile of parts of the host genome, including genes. Double-stranded or highly structured RNAs can trigger the RNA-silencing process, in which they are processed by Dicer-like endonucleases into small interfering RNAs (siRNAs) or microRNAs (miRNAs) of three length classes. According to the currently widely accepted RdDM model, the 21- and 22-nucleotide miRNA and siRNA are incorporated into Argonaute proteins and then direct them to sequence-specific RNAs, leading either to their cleavage or inhibition of translation, or, less commonly, to directing DRM2 to methylate cognate DNA. In contrast, 24-nucleotide siRNAs direct DRM2 to their cognate DNA and are thus involved in RNA-directed DNA methylation (RdDM) [28,29,30].

The estimation of the methylation status of the genome can be performed by several methods, such as chromatography [31], ELISA-based methods [32], AFLP [33], and RFLP [34], whose main advantages are robustness, accessibility, and price. However, the main advantage of HPLC over the other methods is the ability to determine the DNA methylation status on the whole genome level. The disadvantages of ELISA-based methods and AFLP- or RFLP-based methods are high variability and poor resolution of multiple DNA bands, respectively. In most studies focused on identifying specific regions in the plant host genome, researchers have used bisulfite sequencing as this technique is considered the “gold standard” for this purpose. The cost and difficulties in the analysis of NGS data are the only two major limitations [35]. However, it is often used in combination with another method to enrich the sequencing library for specific targets, such as immunoprecipitation, which helps to reduce the complexity of the generated sequencing data and improve the overall cost–benefit ratio. To our knowledge, the level of 5-methylcytosine (5-mC) in the hop genome has never been studied, so the global estimate may serve as a starting point and the results will be useful for further research.

The most convenient method for analysing the genome-wide level of 5-mC is first to hydrolyse the DNA and then analyse the hydrolytic products (nucleotides or nucleobases) by chromatography. The hydrolytic products of DNA can be analysed by various chromatographic techniques, with ion chromatography [36,37,38,39] and reversed-phase chromatography [40,41,42,43,44,45] being the most commonly used techniques. More recently, hydrophilic-interaction chromatography (HILIC) has also been used for this task [46,47]. Acid hydrolysis of DNA [46,48,49,50] is a more favourable option for nucleotide composition analysis as it does not impose limitations, such as DNA size, as in enzymatic hydrolysis [43,45]. Conversely, acid hydrolysis can generate undesirable products such as deamination products [41,49], although such products are usually generated in limited amounts depending on the conditions. Acid hydrolysis can be exhaustive, leading to complete hydrolysis down to bare nucleobases [46]. From an analytical point of view, this is particularly useful in reversed-phase chromatography as the nucleobases are less polar. Therefore, the differences in polarity between compounds are more advantageous from a chromatographic perspective [51] than in the analysis of nucleosides. In HPLC analysis, nucleobases are usually detected with either UV or MS detectors, the latter being more sensitive [47,49]. MS however, has some limitations regarding the composition of the mobile phase. In addition to the mandatory use of a volatile mobile phase, MS detection is improved by ESI ionization when organic modifiers are present in higher percentages [52]. Unfortunately, this is usually not the case even for reversed-phase analysis of nucleobases due to their inherent high polarity [49].

In the present study, we investigate how CBCVd, HLVd, HSVd, and their combinations affect the genome-wide level of cytosine methylation in viroid-infected hop plants by determining the 5-mC level in hop genomic DNA using the HPLC-UV method. To improve our understanding of DNA methylation dynamics in viroid-infected hop plants, we identified, characterized, and phylogenetically analysed hop specific DNA methylases and DNA demethylases, followed by an analysis of their differential gene expression in viroid-infected hop plants.

## 2. Results

### 2.1. Monitoring the Presence of Viroids

Total RNA was extracted from fully developed leaves of the tested hop plants. To confirm the presence or absence of CBCVd, HLVd, and HSVd in the extracted RNA samples, RT-qPCR was performed. Samples were determined to be positive for a particular viroid when amplification occurred (Appendix A). No viroid amplification was detected in the viroid-free hop plants or in the non-template control samples. Melting curve analysis confirmed the presence of specific amplification products and the absence of non-specific amplicons (data not shown). 

### 2.2. Level of Cytosine Methylation

In the present work, we studied the level of cytosine methylation in the genomic DNA of hop plants infected with different viroids, CBCVd, HLVd, and HSVd, and their combinations. In addition, viroid-free hop plants were included in the experiment as a biological control group. The level of cytosine methylation was examined in the extracted DNA of mature leaves using an adapted HPLC-UV method to determine the relative content of 5-mC. Data for individual plants are presented in Appendix A. HPLC-UV analysis of hydrolysed hop genomic DNA revealed that the average 5-mC level in all hop plants tested was 27.1% (*n* = 20). In the viroid-free hop plants (*n* = 3), the average 5-mC level was 26.7%. Furthermore, the level of 5-mC in the viroid-infected hop plants ranged from the lowest value of 23.7% (*n* = 3) in those infected with all three viroids to the highest measured value of 31.4% (*n* = 3) in those infected with HLVd (Figure 1), indicating differential effects of the viroids and their combinations on the methylation level of hop genomic DNA. Duncan’s new multiple range test was performed on the 5-mC data of the hop plants tested and resulted in three groups, A, B, and C (Appendix A), based on the statistical significance (*p*-value < 0.05).

Hop plants single-infected with HLVd, CBCVd, and their coinfection had a higher level of 5-mC compared to the viroid-free hop plants (Figure 1). However, only the HLVd-infected hop plants had a significantly higher 5-mC level than the viroid-free hop plants. On the other hand, hop plants infected with HSVd and hop plants infected simultaneously with CBCVd, HLVd, and HSVd were found to have lower 5-mC than the viroid-free hop plants. Notably, only the hop plants infected with all three viroids had significantly lower 5-mC level compared to the viroid-free hop plants (Figure 1). These observations suggest that HLVd alone has a strong effect leading to hypermethylated genomic DNA, whereas the nature of the effect is reversed and genomic DNA is hypomethylated in hop plants infected with all three viroids. 

### 2.3. Identification and Structural Analysis of DNA Methylase and Demethylase Genes

A set of DNA methylases and demethylases from *A. thaliana* (Appendix A) retrieved from the UniProtKB database were used to identify homologous regions in the hop’s draft genome. Using tBLASTn analysis, we were able to identify four homologous genes, CMT, DME, DNMT, and DRM (Table 1). The gene models were further manually annotated using RNA sequencing data [53] and BLAST comparisons. Subsequently, all identified genes were analysed with Pfam 35.0 to predict their protein domains. The basic characteristics of the genes are summarised in Table 1. The CDS sequences of the gene models were uploaded to NCBI and can be found using the accession number ON863693 for CMT, ON863694 for DME, ON863695 for DNMT, and ON863696 DRM.

The CMT gene identified in the hop’s draft genome has 10 introns and its polypeptide length is 424 amino acids. It has a predicted molecular weight of 50.91 kDa. The best BLASTn hit is a homolog from its closest relative *C. sativa* (XM_030631478.1), with which it shares 94.37% sequence identity but has 15 introns. At the protein level, the hop CMT has 93.41% of sequence identity with a homolog from *C. sativa* (KAF4401783.1). The hop DME gene has 22 introns, is 1827 amino acids long, and has a molecular weight of 204.31 kDa. According to BLASTn search, it has a sequence identity of 84.17% with a homolog from *C. sativa* (XM_030633645.1), which has the same number of introns. The BLASTp search revealed a homolog from *C. sativa* (XP_030489501.1) as the best match for hop DME, with which it has 72.85% sequence identity. The DNMT gene from the hop has 11 introns, a polypeptide length of 1586 amino acids, and a predicted molecular weight of 178.23 kDa. It has 91.57% sequence identity with a homolog from *C. sativa* (XM_030640928.1). Like DNMT from the hop plant, it consists of 11 introns and has 90.76% protein sequence identity with hop DNMT according to the BLASTp search (XP_030496788.1). Finally, the hop DRM gene has three introns and its polypeptide length is 431 amino acids with a predicted molecular weight of 49.37 kDa. A homolog from *C. sativa* (XM_030627066.1) with 92.59% sequence identity and nine introns were found based on the BLASTn search for the hop DRM gene. The best BLASTp result for the hop DRM was a homolog also from *C. sativa* (KAF4381573.1), with which it shares 94.90% sequence identity. A protein domain search using Pfam 35.0 revealed that the hop CMT contains a typical C-5 cytosine-specific DNA methylase domain (PF00145), located after the Chromo (CHRomatin Organisation Modifier) domain (PF00385) (Figure 2). The same domains were found in homologs from *A. thaliana* (Appendix A) and in the homolog from *C. sativa* (KAF4401783.1). However, a BAH domain (PF01426) present in the homologs was not found in the hop CMT. According to Pfam 35.0, the hop DME gene consists of two domains, an RRM in Demeter (PF15628) and a Permuted single zf-CXXC unit (PF15629), the first domain being involved in the release of the methyl group from DNA and thus having demethylase activity (Figure 2). The DNMT identified in hop also has a DNA methylase domain (PF00145). In addition, it consists of two methylase domains, the Cytosine specific DNA methyltransferase replication foci domain (PF12047), which is located before two BAH domains (PF01426) (Figure 2). Finally, only one DNA methylase domain (PF00145) was found in the hop DRM (Figure 2). The same domains were found in DNMT, DME, and DRM homologs from *A. thaliana* (Appendix A) and *C. sativa* (XP_030489501.1, XP_030496788.1, and KAF4381573.1).

### 2.4. Phylogenetic Analysis

Phylogenetic analysis was carried out to determine the relatedness of the genes for hop DNA methylases and demethylases predicted genes within the selected plant species. Polypeptide sequences were retrieved from the UniProtKB database (Appendix A) and aligned using the algorithm MUSCLE, whereupon a phylogenetic tree was constructed for each protein group using the maximum likelihood neighbour-joining method (Figure 3). As anticipated, the identified proteins in hop plants cluster together with proteins from more closely related species of the order Rosales (*C. sativa*, *M. notabilis*, *P. andersonii*, *P. avium*, *P. dulcis*, *P. persica*, *R. chinensis*, *P. ussuriensis* × *P. communis*, *T. orientale*, *Z. jujuba*). The CMT protein sequence found in the hop plant is 93.41% identical to its homolog A0A7J6H953 from *C. sativa*, 86.34% identical to A0A2P5AZ43 from *P. andersonii*, 86.02% identical to A0A2P5FAA8 from *T. orientale*, 80.22% identical to W9SFR4 from *M. notabilis*, and 66.36% identical to Q94F87 from *A. thaliana* (Figure 3a). In addition, the HlDME protein shares 73.38% sequence identity with its homolog A0A803NWS7 from *C. sativa*, 62.18% with A0A2P5EHW2 from *T. orientale*, 46.78% with A0A4Y1RRI4 from *P. dulcis*, 46.85% with A0A251N4F8 from *P. persica*, 48.46% with A0A2P6PPX8 from *R. chinensis*, and 49.44% with Q8LK56 from *A. thaliana* (Figure 3b). The hop DNMT protein has 90.19% protein sequence identity with A0A7J6IA83 from *C. sativa*, 84.45% identity with A0A2P5ALY0 from *T. orientale*, 83.02% with A0A2P5B9G3 from *P. andersonii*, 74.31% with A0A6P3ZL43 from *Z. jujuba*, and only 60.91% with P34881 from *A. thaliana* (Figure 3c). Finally, the DRM protein identified in the hop plant has a protein sequence identity of 94.42% with A0A7J6GF98 from *C. sativa*, 92.79% with A0A2P5BUM3 and A0A2P5EF95 from *P. andersonii* and *T. orientale*, respectively, 77.57% with A0A6P3ZKH9 from *Z. jujuba*, 77.73% with A0A6P5SK83 from *P. avium*, 73.62% with A0A2P6S5M2 from *R. chinensis*, 74.83% with A0A5N5GHK4 from *P. ussuriensis* × *P. communis*, and only 63.43% with Q9M548 from *A. thaliana* (Figure 3d). 

### 2.5. Differential Gene Expression Analysis of DNA (de)Methylase Genes

We designed a RT-qPCR experiment to analyse differential gene expression of the newly identified hop specific genes CMT, DME, DNMT, and DRM in viroid-infected hop plants. RT-qPCR results as relative gene expression levels (Log2 fold-change) are summarized in Appendix A.

Compared to the viroid-free hop plants, the CMT gene showed a non-significant tendency to be down-regulated in all infected hop plants tested (from −0.88 to −0.32 Log2 fold- change, *p*-value > 0.05), except in the plants infected simultaneously with all three viroids, in which it was significantly up-regulated (1.19 Log2 fold-change, *p*-value < 0.05). A similar trend was seen for the DME gene, with a non-significant down-regulated trend in all infected hop plants tested (from −0.96 to −0.27 Log2 fold-change, *p*-value > 0.05), but was again strongly and significantly up-regulated in hop plants infected with all three viroids (1.07 Log2 fold- change, *p*-value < 0.05). In addition, RT-qPCR results (Figure 4) show significant up-regulation of DNMT in hop plants infected with CBCVd (0.34 Log2 fold-change, *p*-value < 0.05), and in hop plants infected with all three viroids (0.38 Log2 fold-change, *p*-value < 0.05), while in hop plants infected with HSVd, the up-regulation was not significant (0.29 Log2 fold-change, *p*-value > 0.05). Moreover, DNMT had a trend of down-regulation in hop plants infected with HLVd (−0.58 Log2 fold-change, *p*-value > 0.05) and in those coinfected with CBCVd and HLVd (−0.40 Log2 fold-change, *p*-value > 0.05). Finally, a further down-regulation trend of the DRM gene was observed in all infected hop plants tested (from −0.60 to −0.28 Log2 fold-change, *p*-value > 0.05), except for plants infected with all three viroids, where it was significantly up-regulated (1.19 Log2 fold-change, *p*-value < 0.05). These results mainly suggest a synergistic effect of all three viroids simultaneously infecting hop plants and thus contributing to an altered expression of hop specific genes.

## 3. Discussion

DNA methylation is one of the three types of epigenetic modifications that regulate gene expression without altering the DNA sequence [1]. The changes in DNA methylation can cause some genes to be silenced or activated, resulting in changes in the phenotype. Similarly, the DNA (de)methylation is an enzymatically driven and highly dynamic process [55]. In the present work, we studied the level of cytosine methylation of hop genomic DNA in viroid-free and viroid-infected hop plants by HPLC-UV. In addition, we performed differential gene expression analysis using RT-qPCR of the newly identified hop genes, CMT, DME, DNMT and DRM.

We adapted a method [50] for HPLC-UV analysis of the 5-mC level in the genomic DNA of hop plants. Formic acid is a less aggressive option for DNA hydrolysis in our case as there are fewer unwanted nucleobase terminations, such as deamination, than with hydrochloric acid [41,49]. The choice of 130 °C for hydrolysis was found to be the least efficient option by Shibayama et al. [50] as longer hydrolysis times are required. On the other hand, the use of a lower hydrolysis temperature, such as 130 °C, resulted in lower pressure within the vial during hydrolysis, and furthermore, almost no vial cap breakage occurred at this temperature, in contrast to higher hydrolysis temperatures (e.g., 150 °C), at which the cap broke or at least leaked in a large percentage of vials used. Very similar hydrolysis conditions were previously used by Sotgia et al. [46] for the analysis of cytosine methylation levels but with a shorter hydrolysis time (80 min). A hydrolysis time of three hours was found to be sufficient to consider DNA hydrolysis complete in terms of the consistency of the results (Appendix A). A small percentage of DNA may remain unhydrolyzed. However, this does not significantly affect the estimation of the 5-mC level, since the relative nucleotide (or nucleobase) composition is required for the determination (Appendix A). The ratios between individual nucleobases were found to be consistently in the range of 90 to 180 min hydrolysis time. Sample preparation in the same vial used for HPLC analysis, without further manipulation, is extremely advantageous in minimising the sample preparation labour and the occurrence of cross-contamination. The disadvantage, of course, is the presence of acid and the lack of a further concentration step, resulting in lower working analyte concentrations. The linearity of the HPLC-UV method was evaluated over a concentration range from 1 µM to 50 µM for each analyte (Appendix A). The calibration curves were set with the intercept at zero, and their slopes and correlation coefficients were calculated. At the lowest calibration point (i.e., 1 µM), the signal-to-noise ratios for cytosine and methylcytosine were 6.4:1 and 15.7:1, respectively. The determination of the limits of detection (LOD) was outside of scope and therefore not assessed, as all working concentrations of the analytes in the sample hydrolysates were well above this, i.e., 3–35 µM and 1.5–16 µM for cytosine and methylcytosine, respectively. Quantification was based on an external standard.

The present work on the cytosine methylation level of whole genomic DNA of the viroid-infected hop plants contributes to the study of the DNA methylation status of host genomes in pathogen-host systems. Based on the HPLC-UV method for determining the 5-mC in hop genomic DNA, we showed for the first time that the cytosine methylation level of viroid- and virus-free hop plants was 26.7% and depends on the infection status of the hop plants. The HPLC method for estimating the content of 5-mC, 5-methyl-2′-deoxycytidine (5mdC) or 5-methyl-2′-deoxycytidine monophosphate (5-mdCMP) has also been used in other plant species. The 5-mC content in *S. olaracea* ranged from 53.74 to 55.75% [56], in *S. tuberosum* from 15.5 to 15.9% [57], and *T. aestivum* from 15 to 30% [58]. The content of 5-mdC in *E. senticosus* ranged from approximately 10 to 22% [59] and the content of 5-mdCMP in *T. chiensis* was reported to range from 9.1 to 16.7% [43]. The level of cytosine methylation varies greatly between species, as shown by bisulfite sequencing results. The level of DNA methylation of the GC context was lowest in *A. thaliana* at 30.5% and the highest in *B. vulgaris* at 92.5% however, 75% mGC was observed in *C. sativa*, the closest relative of hop plant [7]. 

Our results also show that the DNA methylation level of the hop genome varies depending on the viroid and the combination of viroids infecting hop plants. Indeed, HLVd-infected hop plants had a statistically significant higher cytosine methylation level, suggesting that HLVd infection increases the host cytosine methylation level (Figure 1). Interestingly, HLVd is known to be the least devastating viroid in hop cultivation and is considered a latent pathogen, although its impact on alpha acid production has been reported [60]. Potato spindle tuber viroid (PSTVd) infection has also been associated with hypermethylation of certain DNA sequences in *N. benthamiana* [24], and in potatoes [25]. On the other hand, hop plants infected with CBCVd, HLVd, and HSVd had a statistically significant lower cytosine methylation level, indicating an opposite effect to HLVd infection, leading to a hypomethylated status of hop genomic DNA. Other studies have shown that HSVd infection leads to hypomethylation of ribosomal DNA in cucumber [22,26] and *N. benthamiana* [23]. In our experiment, cytosine methylation levels in CBCVd- or HSVd-infected plants and in CBCVd- and HLVd-coinfected plants were apparently not affected by the pathogens or their combination. However, hop plants infected with CBCVd and HLVd had higher 5-mC levels than CBCVd-infected hop plants and lower 5-mC levels than HLVd-infected hop plants, suggesting an interaction between the two viroids. It should be noted that the level of cytosine methylation was studied at the whole-genome level, implying that differentially hypermethylated and hypomethylated DNA regions may still be present, although if they are present in the same ratio, no difference in total cytosine methylation is detected by the method used. This would be confirmed by an alternative approach to examine the differentially methylated DNA regions, such as bisulfite sequencing. However, viroids have been reported to interact in an antagonistic manner [61], so it is conceivable that CBCVd and/or HSVd interact with HLVd such that cytosine methylation does not ultimately change. 

In plants, DNA is de novo methylated via an RNA-directed DNA methylation pathway [62]. The current RdDM model involves a DRM2, Domains Rearranged Methyltransferase 2, which transfers the methyl group to cognate DNA in the final step. In our study, we included other known DNA methylases and demethylases from *A. thaliana* (Appendix A) in addition to DRM2 and were able to identify three DNA methylases, CMT, DNMT, and DRM, genes and one demethylase, DME, gene in hop. We used Pfam 35.0 to predict the typical protein domains for the gene models created. All hop DNA methylases, CMT, DNMT, and DRM consist of a typical methylase domain (PF00145) and DNA demethylase has a typical RRM in DEMETER domain (PF15628), known for its glycosylase mechanism, through which it removes the methyl group from the cytosine [63,64] (Figure 2). These protein domains have been confirmed in other studies on DNA methylases and DNA demethylases [65,66]. In addition, our phylogenetic analysis, based on a group of proteins from selected plant species from Rosids clade (NCBI: taxid 71275) (Appendix A) confirmed the relatedness of DNA methylases and demethylases from hop plants to orthologs from closely related plant species. The proteins identified in hop formed clusters together with *C. sativa*, *T. orientale*, *P. andersonii*, *P**. avium*, *P. dulcis*, *P. persica*, *R. chinensis*, *P. ussuriensis* × *P. communis*, and *Z. jujuba* (Figure 3). The highest polypeptide sequence similarity between hop plants and the selected plant species (Appendix A) was found for DRM (from 63.43% to 94.42%), followed by CMT (from 66.36% to 93.41%), DNMT (from 60.91% to 90.19%), and DME (from 49.44% to 73.38%). 

Down-regulation of the CMT gene was observed in all viroid-infected hop plants examined, except in plants infected simultaneously with all three viroids, where expression was significantly higher compared to the viroid-free hop plants (Figure 4). This suggests a synergistic relationship between CBCVd, HLVd, and HSVd leading to increased CMT expression. CMT is a conserved plant-specific chromomethylase 3 required for maintaining CHG methylation [67], but its mechanism does not rely on small interfering RNAs (siRNAs) [30]. Similarly, RT-qPCR results showed a tendency for down-regulation of the DME gene in hop plants single-infected with CBCVd, HLVd, or HSVd, and in co-infected plants with CBCVd and HLVd. In contrast, DME gene expression was significantly increased in hop plants infected with all three viroids (Figure 4), again indicating an effect of mixed viroid infection. Increased expression of a DNA demethylase, whose function is to remove the methyl group from DNA [64,68,69], could lead to hypomethylated genomic DNA, as was the case in hop plants infected with all three viroids (Figure 1). Furthermore, our RT-qPCR results revealed a different expression trend for the hop DNMT gene. In fact, DNMT was upregulated in hop plants with a single HSVd or CBCVd infection, respectively, and in hop plants infected with all three viroids. However, significant up-regulation was observed only for the latter two (Figure 4), implying that infection of hop plants with CBCVd or all three viroids results in higher expression of DNMT. Like CMT, DNMT is a conserved methyltransferase known to maintain CG methylation in absence of dsRNA or siRNA [70]. Last but not least, DRM was down-regulated across all viroid-infected hop plants tested, except in those infected with all three viroids, further suggesting a synergistic effect of the co-infected viroids. Synergism of mixed viroid infection resulting in development of disease symptoms has been observed in *C. medica* [71], and *A. citroides* [72]. In particular, the synergistic effect of multiple viroid infections has been shown to alter gene expression [73] and the mRNA surveillance and ribosome biogenesis pathway [53] in hop plants. Moreover, stress-induced hypomethylated DNA has been associated with increased gene expression [9], such as Dicer-like 4 (DCL4) gene [74], and furthermore, DNA hypomethylation at selected pericentromeric regions has been linked to genome-wide priming of defence-related genes [75]. Moreover, PSTVd infection of the common tomato positively affected the expression of genes related to RdDM [27], some of which were also identified in the present study. The observed increase in CMT, DME, and DNMT gene expression in hop plants infected with all three viroids and the increased DNMT gene levels in CBCVd-infected hop plants suggest that viroid infection also affects factors whose mechanism does not depend on siRNAs. Interestingly, infection of hop plants with HLVd leads to hypermethylation of genomic DNA (Figure 1), but on the other hand, all DNA methylases showed a minor, non-significant down-regulation trend (Figure 4), possibly indicating a contribution of other factors to the DNA methylation landscape in HLVd-infected hop plants. Moreover, 5-mC levels were much higher in hop plants infected with HLVd than in hop plants infected with CBCVd, HLVd, and HSVd (Figure 1), in which all three DNA methylases were up-regulated, further implying the overwhelming DNA demethylase activity.

Further studies are needed to identify the DNA regions of the hop genome where cytosine methylation is altered due to viroid infection. Together with a comprehensive study of the hop transcriptome response triggered by viroid infection [53,76], this would help to clarify which genes relevant to hop pathogenicity alter their expression as a consequence of differential DNA methylation. In addition, studying the interactions between vd-siRNAs and host factors, particularly those involved in RdDM, would further confirm the putative influence of viroids on DNA methylation dynamics in hop plants.

## 4. Materials and Methods

### 4.1. Hop Inoculation Experiment 

Virus- and viroid-free hop plants of the clonally propagated cultivar ‘Celeia’ were obtained from the hop nursery of the Institute of Hop Research and Brewing, Žalec, Slovenia. Ten plants for each viroid treatment (CBCVd, HLVd, HSVd, CBCVd and HLVd, CBCVd, HLVd, and HSVd) were biolistically inoculated with 360 ng of dimeric viroid constructs (GenBank X07397, GenBank KM211546, GenBank X07405) using the Helios GeneGun (Bio-Rad Laboratories, Inc., Hercules, CA, USA) [53,73]. Plants were placed in polyethylene bags to maintain high humidity and prevent drying of injured tissues, and these were placed in a growth chamber (25 °C and 16 h of illumination). After one week, the plants were re-potted into 4-litre pots, moved to an isolated field plot, and kept in an insect-proof netting (1.6 mm × 1.6 mm). Virus-free and viroid-free hop plants (controls) were treated under the same conditions. Fully developed leaves from control plants and from viroid-infected plants were sampled in phenological stage BBCH 38 from the same sampling point (4–5 nodes below the apical bud). The biolistically infected plants were in the infection stage 50 mpi (months post inoculation). Infected hop plants from the field were estimated to be infected for at least 2 years, based on visual observations of the infected hop field (data not shown). Leaves were immediately frozen in liquid nitrogen and then pulverized in the laboratory and stored at −80 °C. In subsequent analyses, at least three biological replicates were used for each experimental group. 

### 4.2. HPLC-UV Assay of 5-mC in Hop Genomic DNA 

DNA was extracted using the CTAB method [77]. In the final step, the air-dried pellet was re-suspended in 50 µL TE buffer (10 mM Tris, pH = 8, 1 mM EDTA). Twenty microliters of the DNA solution in TE (concentration between 200 and 1700 ng/µL) was transferred to a crimp-type HPLC vial with a 300 µL glass insert. Sixty microliters of concentrated formic acid were added. The vial was crimped and vortexed. DNA hydrolysis was performed by heating the HPLC vials at 130 °C for 3 h in a drying oven. After heating and cooling, 100 µL of water was added to each vial through the septum using a glass syringe and mixed by vortexing. The resulting sample hydrolysate solutions were injected into the HPLC. A Vanquish (Thermo Fisher Scientific Inc., Waltham, MA, USA) HPLC system equipped with a UV-VIS diode array detector was used. The HPLC column was a Hypercarb (Thermo Fisher Scientific Inc., Waltham, MA, USA), with dimensions of 50 mm × 2.1 mm i.d., a 3 µm particle size and a temperature of 40 °C. The autosampler flush solvent was 10% methanol (*v*/*v*). The wavelength of UV detection was 295 nm. The flow rate was 0.2 mL/min; the injection volume was 1 µL. Mobile phase A consisted of water with 0.1% formic acid (*v*/*v*), and mobile phase B was acetonitrile with 0.1% formic acid (*v*/*v*). The elution gradient was as follows: 0–14 min, 0% B to 35% B; 14–14.1 min, 35% B to 0% B; 14.1–20 min, 0% B. The run time was 20 min. Calibration was performed by injecting an equimolar solution of cytosine and 5-mC, each containing 50 µM in 10% methanol (*v*/*v*), into the HPLC. By determining the ratio between the peak areas, the peak area of 5-mC was normalized to the peak area of cytosine. This normalization factor was used throughout the sample analyses (for each analytical batch) to obtain the relative content of 5-mC in the sample solutions. Data for 5-mC levels were tested for statistical significance across hop plants tested by applying Duncan’s multiple range test [78].

### 4.3. Identification of DNA Methylases and Demethylases 

For the identification of DNA methylases and demethylases, we downloaded a set of protein sequences of *A. thaliana* from UniProtKB (Appendix A) (https://www.uniprot.org/; accessed on 8 March 2022). Unless otherwise indicated, bioinformatics steps were performed using CLC Genomics Workbench (22.0) (QIAGEN, Digital Insight, Aarhus, Denmark). Briefly, homologous sequences were identified by blasting (tBLASTn, E < 0.001) the protein sequences of DNA methylases and demethylases from *A. thaliana* against the hop’s draft genome Cascade [79]. To improve the genetic annotation of the identified sequences in the hop’s draft genome, we used the Large gap read mapping tool (2.0) with default parameters and RNA-sequence data generated in a previous study by our research group [53]. The raw NGS data are publicly available under BioProject number PRJNA342762, BioSample SAMN05767836, SRA run SRR4242068. The protein coding sequence within transcripts was predicted using NCBI ORFfinder (https://www.ncbi.nlm.nih.gov/orffinder/; accessed on 8 March 2022). We then used Pfam 35.0 (http://pfam.xfam.org/; accessed on 8 March 2022) to verify that the protein sequences obtained for DNA methylases and demethylases contain the typical DNA methylase domain (PF00145), and DNA demethylase domain (PF15628), respectively. The theoretical isoelectric point and molecular weight were calculated using the ExPASy Compute pI/Mw tool (https://www.expasy.org/resources/compute-pI-mw; accessed on 8 March 2022). Subsequently, the identified and manually curated gene sequences were blasted against the assembled hop transcriptome of cultivar ‘Celeia’ [80] (BLASTn; E < 0.001) to find the contigs representing transcripts for DNA methylases and demethylases genes, supporting the design of qPCR primers.

### 4.4. Phylogenetic Analysis 

The polypeptide sequences of DNA methylases and demethylases of selected plant species (Appendix A) from the Rosids clade (NCBI: taxid: 71275), including the sequences of the newly identified DNA methylases and demethylase of the hop plant, were used for phylogenetic analysis. Multiple sequence alignment was performed using the algorithm MUSCLE with default parameters implemented in CLC Genomics Workbench (22.0). The three groups of DNA methylases and one group of DNA demethylases were aligned together with their homologs in four separate alignments. Alignments were manually curated by truncating all positions with gaps and missing data, followed by a construction of phylogenetic trees using the maximum likelihood method based on the WAG protein substitution model using default parameters implemented in CLC Genomics Workbench (22.0). The reliability of the tree nodes was tested with 1000 bootstrap replicates.

### 4.5. Total RNA Extraction and qPCR Analysis

RT-qPCR was used to confirm and monitor viroid infection and measure gene expression levels. Total RNA was extracted from the sampled material using the Monarch Total RNA Miniprep Kit (New England Biolabs Inc., Ipswich, MA, USA) according to the manufacturer’s instructions. The concentration and quality of the isolated RNA were checked by NanoVue spectrophotometer (GE Healthcare, Chicago, IL, USA) and Agilent Bioanalyzer electrophoresis using RNA 6000 Nano Kit (Agilent Technologies, Inc., Santa Clara, CA, USA) before further analysis. The cDNA of the viroids was synthesized by first denaturing the secondary structures of the viroids by incubating 500 ng of the isolated RNA at 90 °C for 3 min and then immediately placing it on ice. The same amount of RNA was used for gene expression analysis, but the denaturation step was omitted. For cDNA synthesis, the High-Capacity cDNA Reverse Transcription Kit (Thermo Fisher Scientific Inc., Waltham, MA, USA), was used, following the manufacturer’s instructions. qPCR primers for viroid monitoring and a housekeeping gene, DRH1, were previously developed [73,81], whereas primers for gene expression analysis were designed using Primer3plus (https://www.bioinformatics.nl/cgi-bin/primer3plus/primer3plus.cgi; accessed on 8 March 2022) (Appendix A). The qPCR reactions were performed using the QuantStudio™ 5 Real-Time PCR System (Thermo Fisher Scientific Inc., Waltham, MA, USA). For viroid detection, 1 µL of diluted cDNA (1:100) in nuclease-free water (Integrated DNA Technologies Inc., Coralville, IA, USA) was amplified in a 6 µL reaction using 1X Fast SYBR™ Green Master Mix (Thermo Fisher Scientific Inc., Waltham, MA, USA) and 200 nM primers, while 1 µL of undiluted cDNA and 300 nM primers were used for gene expression analysis. The amplification protocol was as follows: 95 °C for 20 s, followed by 40 cycles of 95 °C for 15 s and 60 °C for 30 s; it ended with a melting curve analysis that included steps of 95 °C for 15 s, then the temperature was dropped to 60 °C for 1 min at a rate of 1.76 °C/s and finally increased to 95 °C for 15 s at a rate of 0.075 °C/s. Relative gene expression levels were calculated according to the Pfaffl method [82] and then tested for statistical significance (*p*-value < 0.05) by applying Duncan’s multiple range test [78] in R Commander [83].

## 5. Conclusions

In summary, the results presented in this work show that the cytosine methylation level of the genomic DNA of the hop plant (*Humulus lupulus* var. ‘Celeia’) can be determined by the HPLC-UV method. Moreover, our study shows for the first time that the level of cytosine methylation in hop plants at BBCH 38 stage is 26.7%. The level of 5-mC varies among the hop plants infected with different viroids and their combinations, but is significantly higher in HLVd-infected hop plants infected and significantly lower in hop plants infected with CBCVd, HLVd, and HSVd than in the viroid-free hop plants. In the present study, we also identified three DNA methylases, CMT, DNMT, DRM, and one DNA demethylase, DME, in hop’s draft genome. Our RT-qPCR results demonstrate a synergistic effect of the viroids infecting the hop plants with respect to differential expression of the newly identified genes. This indicates that the combination of CBCVd, HLVd, and HSVd affects an important factor, DRM, of RdDM pathway in the infected hop plants. In addition, the same combination of viroids also contributes to the upregulation of the DME gene. Finally, infection of hop plants with either CBCVd, CBCVd, HLVd or HSVd leads to an upregulation of the DNMT gene, which maintains the DNA methylation patterns and is independent of the small interfering RNAs.

## Figures and Tables

**Figure 1 cells-11-02592-f001:**
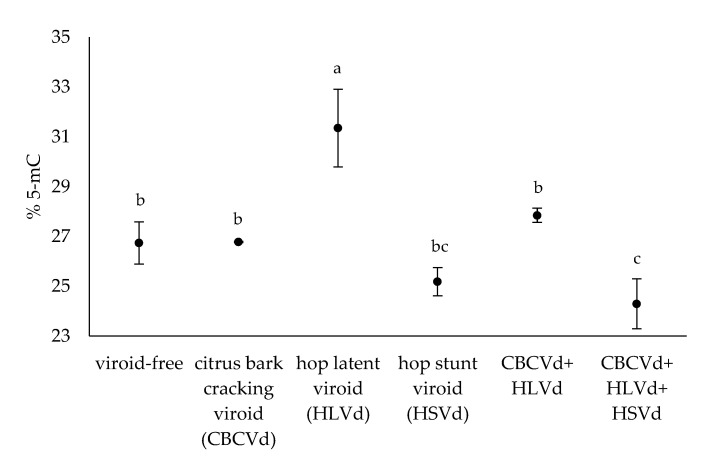
The average % 5-mC in the hop plants tested. The letters indicate statistically significant results (*p*-value < 0.05) and the bars show the standard errors.

**Figure 2 cells-11-02592-f002:**
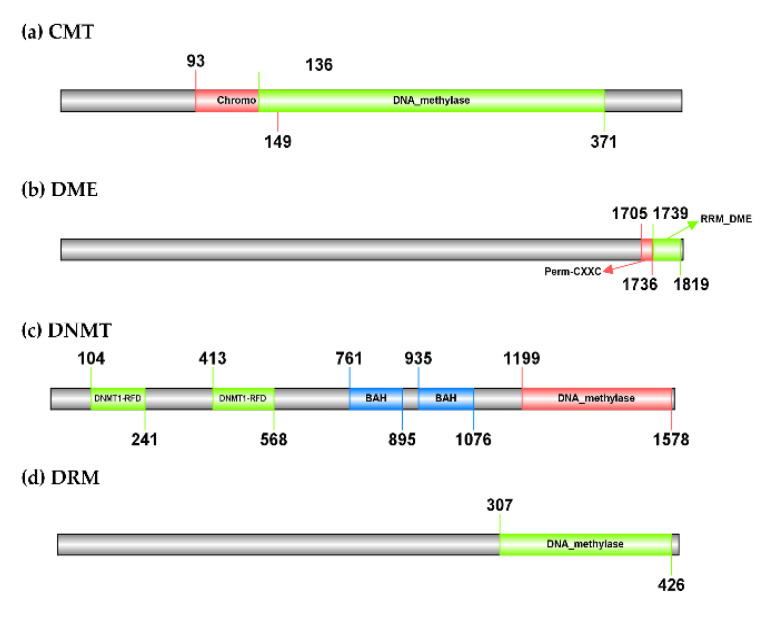
Organisation of the domains of the three identified DNA methylases and one demethylase in *Humulus lupulus*. A typical C-5 cytosine-specific DNA methylase domain (PF00145) was found in CMT, DNMT, and DRM genes. In addition, CMT contains one (CHRromatin Organization MOdifier) domain (PF00385) and DNMT contains two consecutive cytosine-specific DNA methyltransferase replication foci domains (PF12047), before the two BAH domains (PF01426). The DME gene consists of two domains, an RRM in demeter (PF15628) involved in the release of methyl groups from DNA and a permuted single zf-CXXC unit (PF15629). The online tool Illustrator of Biological Sequences (IBS; http://ibs.biocuckoo.org/online.php; accessed on 8 March 2022) was used to visualize the protein domains.

**Figure 3 cells-11-02592-f003:**
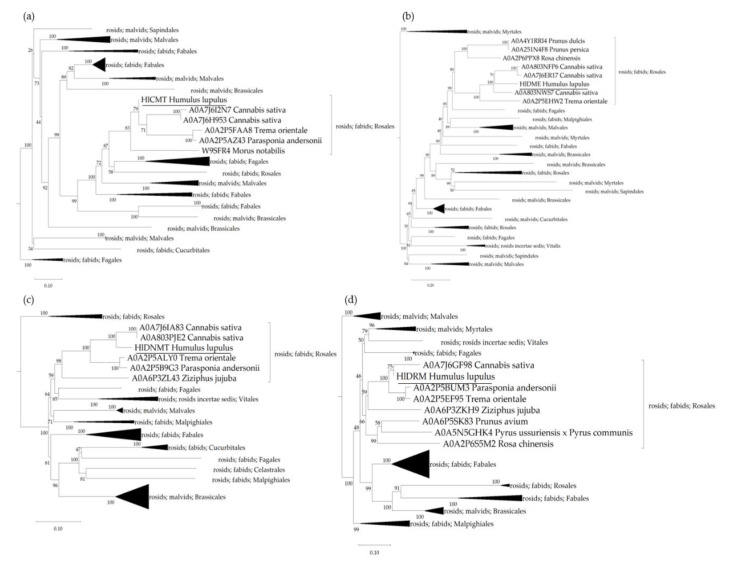
Phylogenetic trees of the Chromomethylase (CMT), Demethylase (DME), DNA methyltransferase (DNMT), and Domains Rearranged Methyltransferase (DRM) proteins for the selected plant species (Appendix A), containing the newly identified DNA methylases and demethylases in hop. The polypeptide sequences were aligned using the MUSCLE algorithm, and the maximum likelihood phylogeny based on the WAG protein substitution model was used to construct the neighbour-joining trees for the (**a**) CMT, (**b**) DME, (**c**) DNMT, and (**d**) DRM protein groups. The numbers above the nodes indicate the reliability of 1000 bootstrap replicates. Scale bars represent amino acid substitutions per site. The software MEGA11 was used for the visualisation of the phylogenetic trees [54].

**Figure 4 cells-11-02592-f004:**
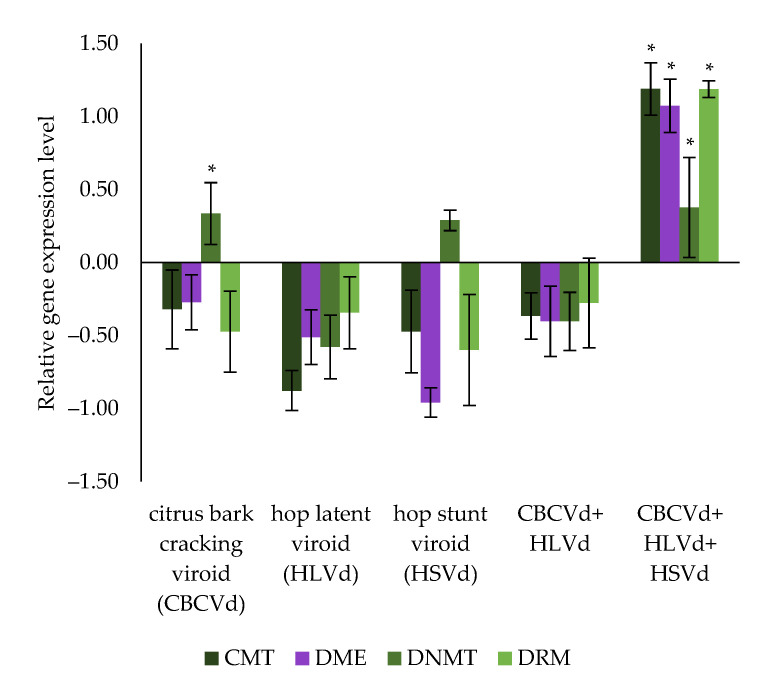
Differential gene expression of the newly identified CMT, DME, DNTM, and DRM in hop’s draft genome, across hop plants infected with different viroids and their combination (CBCVd, HLVd, HSVd, CBCVd and HLVd, and CBCVd, HLVd, and HSVd) compared with viroid-free hop plants. An asterisk indicates a statistically significant result (*p*-value < 0.05), and the bars show the standard errors.

**Table 1 cells-11-02592-t001:** Characteristics of the identified DNA methylases and demethylases in the hop’s draft genome.

Gene	ORF Length (nt)	Number of Introns	Length (aa)	Mw (kDa)	pI	Gene Model	Start	End	BLASTp Best Hit
CMT	1272	10	424	50.91	6.84	001524F.g1	3569	8263	KAF4401783.1
DME	5484	22	1827	204.31	6.54	001841F.g3	65831	76075	XP_030489501.1
DNMT	4761	11	1586	178.23	5.91	000387F.g29	809291	815800	XP_030496788.1
DRM	1296	3	431	49.37	8.66	000004F.g82	2354798	2357141	KAF4381573.1

pI stands for the isoelectric point of a protein; Mw stands for the molecular weight of a protein in kilodaltons (kDa).

## Data Availability

Data supporting results are within the paper and its Appendix A. The computationally assembled hops transcriptome is available from our research group and the raw NGS sequences of hops transcriptome are deposited in NCBI’s SRA archive under BioProject number PRJNA342762, BioSample SAMN05767836, SRA run SRR4242068: https://www.ncbi.nlm.nih.gov/sra/?term=SRR4242068 (accessed on 8 March 2022). The hop draft genome was obtained from the HopBase genomic resource which is available at http://hopbase.org (accessed on 8 March 2022) and http://hopbase.cgrb.oregonstate.edu (accessed on 8 March 2022). All the data are presented in the manuscript.

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
