# Peer review of "Cytosine Methylation in Genomic DNA and Characterization of DNA Methylases and Demethylases and Their Expression Profiles in Viroid-Infected Hop Plants (Humulus lupulus Var. ‘Celeia’)"

_cells, 2022, doi:10.3390/cells11162592_

Round 1

Reviewer 1 Report

The manuscript by Secnik and collaborators shows the interplay between DNA methylation and viroid infection in hop plants (Humulus lupulus). A series of pioneering articles 2014-2016 (references 22 to 27 in this manuscript) showed alterations in the methylation pattern of some genomic regions of the host plants in viroid infection, establishing that interplay with host methylation may be important in host-viroid interaction and symptom development. However, this aspect of viroid biology has not received so far the deserved interest. This manuscript by Secnik stablishes the foundation to study host methylation in a really interesting viroid pathosystem consisting in hop plants infected by up to three different viroids or co-infected by combinations of two of them, even the three of them. 

I consider the manuscript highly interesting for researchers in the viroid community and also for researchers involved in methylation of plant genes. Basically, authors quantified methylation level (5-methylcytosine) of mock and viroid-infected hop plants, characterized DNA methylases and demethylases in hop, and analyzed expression change of these genes upon viroid infection. Although this manuscript may be considered an initial step in viroid-related methylation research, since further studies should focus in particular host genes or regions of the genome, the novelty of the results guarantee publication. Of note, authors optimize a relatively straight-forward protocol to quantify DNA methylation by HPLC-UV. Also remarkable, the viroid infection of hop plants has been revealed as a really important disease outbreak in central Europe, which demands efforts both in basic and applied research by the viroid research community. 

Experiments have been well planned and, to my understanding, well performed. Results are really interesting. The manuscript, maybe a little bit too lengthy in some aspects, is however very well written; so, I would not suggest reducing. These are a few comments to improve the manuscript: 

1. Page 2, 2nd paragraph, follow recent recommendation to name viroid species. Since authors are not referring to viroids in a taxonomic context, my suggestion is not to capitalize first letter. 

2. Page 3, 1st paragraph, the word nucleobase is a little awkward, at least for this reviewer. I suggest to keep in the more standard nucleotide or nucleoside, depending whether the monomer keeps or not the phosphate group. 

3. Introduction and Discussion are lengthy in some aspects, although they read well. However, some description and discussion is missing about pioneering work on host genome methylation upon viroid infection (references 22 to 27). 

4. Check reference style for 4, 31, 71 and 78.

Reviewer 2 Report

DNA methylation is an important factor mediating the plant response to biotic stress. Virus- and viroid infections can cause changes in the level of DNA methylation of both the host and the infectious agent, and affect gene expression downstream. In the presented paper, an adapted HPLC-UV method was used for the first time to estimate the level of 5-methylcytosine  in hops that can be applied to study the DNA methylation status of other hosts genomes in pathogen-host systems. The study identified for the first time the genes encoding DNA methylating and demethylating enzymes in hops and examined their expression at the transcriptional level in uninfected and viroid pathogen-infected plants. However, further studies are needed to identify the specific DNA regions of the hop genome whose methylation status changes during viroid infection, the results obtained are useful for elucidating the regulatory pathways during viroid infection.

1.     In Materials and methods, it is stated that 360 ng of each viroid construct was administered during inoculation. This means that the total viroid load in plants infected with a combination of 3 viroid species (CBCVd, HLVd and HSVd) is approximately 3 times greater than that of plants infected with only 1 viroid species. Supplementary Table 1 also shows the viroid load when the leaf samples were collected for analysis, which confirmed the higher total pathogen load in plants infected with the triple combination compared to the single viroid infection. 

In this regard, could the observed differences in gene expression between the single, dual, and triple treatments be expected to reflect not only a synergistic effect of the viroid combination but also be a consequence of the higher viroid load overall?

2.     What is the reason for choosing the phenological stage BBCH 38 for analysis?

3.     How the gene IDs shown in section 2.3. (lane 158) relate to the gene models in Table 2 and the hope genome?

4.     Regarding DME, two domains are represented in the protein structure - Perm-CXXC and RRM_DME in Figure 2, but the third domain ENDO3c which is specific for this enzyme and can be found in the most closely related ortholog in C. sativa, is not found in the hop. How do the authors interpret the absence of this domain in hop DME? Can this result in altered enzyme activity?

5.     In Conclusions, the authors state that ”Moreover, our study shows for the first time that the 546 level of cytosine methylation in hop plants is 26.7%.” It should be specified that this observation was made at a certain stage of development.
